# Versatile DMA Engine for High-Energy Physics Data Acquisition Implemented with High-Level Synthesis

**Wojciech Marek Zabołotny** 

Faculty of Electronics and Information Technology, Institute of Electronic Systems, Warsaw University of Technology, Nowowiejska 15/19, 00-65 Warszawa, Poland; wojciech.zabolotny@pw.edu.pl

**Abstract:** FPGA-based cards for data concentration and readout are often used in data acquisition (DAQ) systems for high-energy physics experiments. The DMA engines implemented in FPGA enable efficient data transfer to the processing system's memory. This paper presents a versatile DMA engine. It may be used in systems with FPGA-equipped PCIe boards hosted in a server and MPSoC-based systems with programmable logic connected directly to the AXI system bus. The core part of the engine is implemented in HLS to simplify further development and modifications. The design is modular and may be easily integrated with the user's DAQ logic, assuming it delivers the data via a standard AXI-Stream interface. The engine and accompanying software are designed with flexibility in mind. They offer a simple single-packet mode for debugging and a high-performance multi-packet mode fully utilizing the computational power of the processing system. The number of used DAQ cards and the amount of memory used for the DMA buffer may be modified in the runtime without rebooting the system. That is particularly useful in the development and test setups. This paper also presents the development and testing methodology. The whole design is open-source and available in public repositories.

**Keywords:** FPGA; DMA; HEP; DAQ; HLS

## 1. Introduction

In data acquisition systems (DAQ) for high energy physics (HEP) experiments, the significant volume of data from the front end electronics (FEE) must be collected, submitted to preprocessing, and transferred to the computer systems or whole grids responsible for final processing and archiving.

Connection to the FEE often uses non-standard high-speed interfaces (Examples of such non-standard high-speed interfaces may be GBT [1,2], and lpGBT [3,4] links widely used for connecting FEE in CERN experiments). Data preprocessing is usually associated with being highly parallel and fast, but with simple calculations on data received from numerous measurement channels.

Therefore, this section of DAQ is usually implemented using the field programmable gate array (FPGA) chips, which give the additional advantageous possibility of upgrading the communication and processing algorithms throughout the whole experiment's lifetime.

Finally, the preprocessed data must be concentrated and written to the computer system's memory in a form enabling efficient final processing. That task should be accomplished via direct memory access (DMA) to avoid wasting CPU time on simple data transfer. If the FPGA responsible for the reception and processing data has direct access to the system bus of the computer system, implementing the necessary DMA engine in that FPGA enables efficient data handling and flexibility regarding the layout of data stored in the memory.

The data acquisition in HEP experiments may run continuously for a long time (many hours or even days). Therefore, the DMA engine must be capable of performing the

acquisition in the continuous mode. The memory buffer for storing the acquired data must be a circular buffer, and protections against buffer overflow must be implemented.

However, implementing complex algorithms in FPGA is a relatively complex task requiring highly skilled engineers. Full use of the flexibility provided by FPGA can be significantly facilitated if programmers without this expertise can be involved in creating or modifying such algorithms. That may be possible with high-level synthesis (HLS)—a technology enabling the automated conversion of algorithms written in C/C++ into an FPGA implementation. HLS is successfully and widely used in data processing, but less often in designing control blocks and hardware interfaces. This paper describes using HLS to implement a simple yet efficient and flexible DMA engine for HEP-oriented DAQ systems. The system requirements have been formulated based on reviewing the hardware and software platform considerations in Sections 1.1 and 1.2. Section 2 describes the existing solutions. Based on those requirements and prior art, Section 3 describes the concept of a novelty DMA system. Its original features are:

- The core of the DMA engine is implemented in HLS. This enables easy modification of the data handling.
- The DMA engine (the FPGA design and the kernel driver) is compatible with FPGAs connected via PCI-Express in servers and MPSoC chips.
- The scatter–gather DMA buffer based on huge pages enables flexible memory management even with a standard distribution Linux kernel.
- The solution utilizes the QEMU model of the DMA engine implemented in C. That enables efficient development and testing of the device driver and application in the virtual environment [5].

### 1.1. Hardware Platform Considerations

Enabling a FPGA to fully control the DMA data transfer to the host computer's memory requires a tight connection between the FPGA and the system bus. Currently, there are two hardware solutions commonly used for that. The first is used in SoC (System on Chip) or MPSoC (Multi-Processor System on Chip) chips, where the programmable logic (PL) is connected with the processing system (PS) in the same integrated circuit via one or more AXI buses. The digital system implemented in PL may contain not only AXI Slaves, but also AXI Masters, enabling the creation of the DMA engine.

The second solution uses FPGA-based extension cards connected to the computer system's PCI-Express (PCIe) interface. In this solution, as in the previous one, the FPGA may implement not only bus slaves, but also bus masters, which can be used as a DMA engine.

In an MPSoC or SoC system, the DMA engine directly controls the system's AXI bus. In the PCIe-based solution, the AXI bus is provided by the AXI–PCIe bridge [6,7]. That bridge translates the transactions performed by the DMA engine on the AXI bus into the equivalent transactions on the host's PCIe bus. Thanks to that, both hardware configurations may work with the same DMA engine.

The next necessary design choice is the selection of the input interface for the DMA engine. For further processing in the computer system, it is beneficial that the preprocessed and concentrated data are split into smaller portions (packets) supplemented with additional metadata, which may describe their origin, time of acquisition, and other information necessary for the particular experiment. A natural solution for transmitting data structured in that way inside of FPGA is the AXI-Stream [8] interface.

The concept of a versatile DMA engine based on the above considerations is shown in Figure 1.

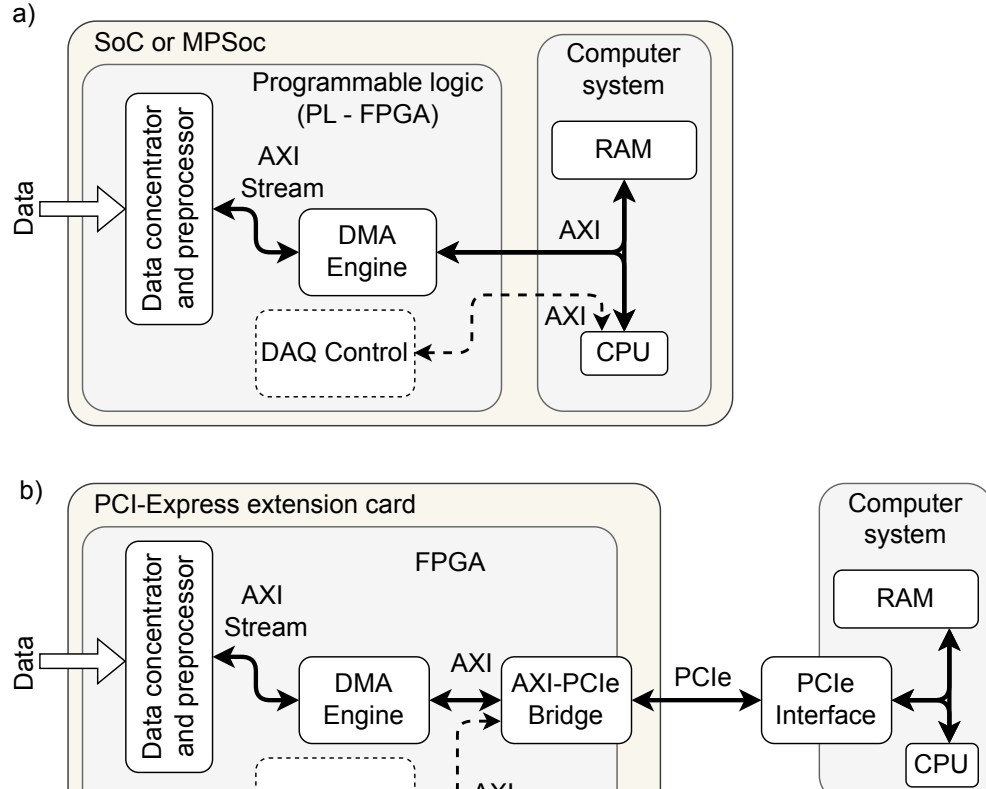

**Figure 1.** Two typical hardware architectures using the DMA engine implemented in FPGA; (**a**) based on MPSoC with FPGA and processing system connected directly via AXI interface, (**b**) based on a FPGA-equipped PCIe extension card. The AXI-PCIe bridge enables using the same DMA engine as in case (**a**). The "DAQ Control" block represents other user-defined logic in FPGA used to control the DAQ system. It is connected to a separate AXI bus and accessible as another character device in Linux (see Section 5).

### 1.2. Computer System Considerations

The operating system typically used in HEP DAQ nodes is Linux. In the hardware platforms mentioned in the previous section, Linux uses the *virtual memory* implemented with *paging* [9]. That simplifies memory management by eliminating problems resulting from memory fragmentation, enables memory protection, and allows running each application in its own virtual address space. However, from the DMA point of view, the *paging* significantly complicates the operation of DMA engines. The kernel and application's virtual memory addresses are not identical to the physical addresses. They are translated using the *page tables* and *page directories* (see Figure 2).

Due to the memory fragmentation during the computer system's operation, allocating a large physically contiguous buffer may be difficult. The large memory buffer, contiguous in the virtual address space, may consist of multiple smaller contiguous buffers (in the worst case, just single *memory pages*) scattered in the physical address space. Such a buffer is called a scatter–gather buffer (in short, SG buffer).

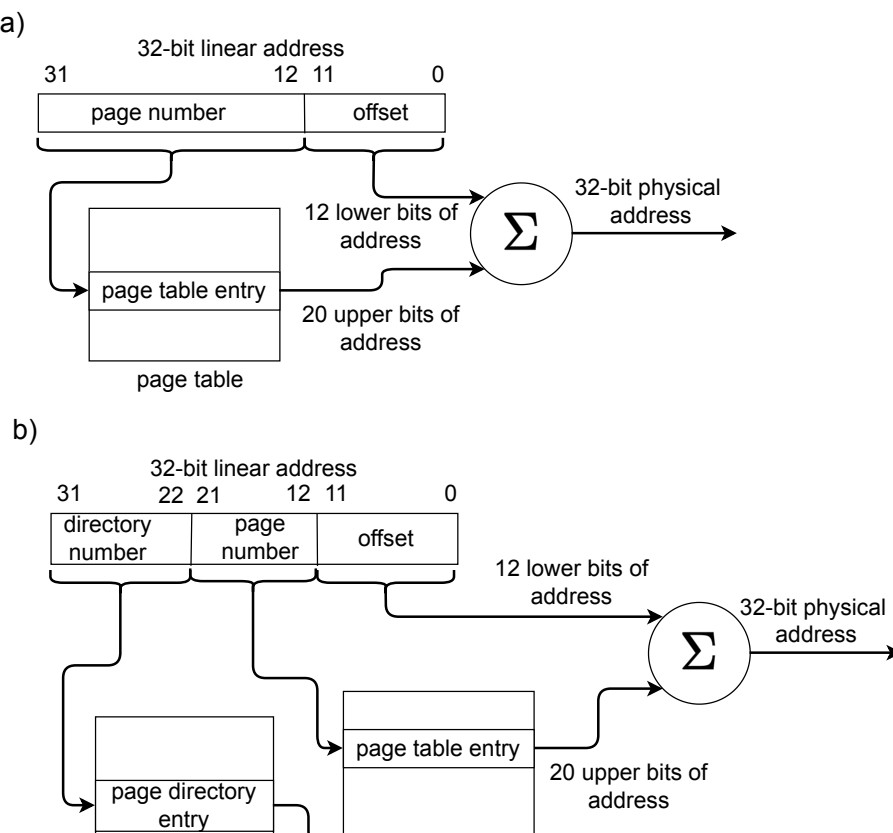

**Figure 2.** Translation of linear (virtual) addresses into physical addresses with paging (in a 32-bit system for simplicity). (**a**) The simple one-level paging. The 20 upper bits of the virtual address are the page number. The page's physical address is stored in the corresponding page table entry. The lower 12 bits of the address are the offset inside the page. That scheme is rarely used because it requires creating large page tables (with $2^{20}$ entries for the whole 32-bit address space). (**b**) The two-level paging. The bits 31–22 are the page table number. The address of the associated page table is taken from the page directory entry. There may be an empty directory entry for unused address areas, and the associated page table is not created. That solves the problem of large page tables. The bits 21–12 are the page number in the page table. The lowest 12 bits are the offset in the page. The figure is based on the documentation published by Intel [10].

The DMA engine does not use virtual addresses. It uses the physical addresses or the bus addresses additionally translated by the bus bridge. If the host computer system is equipped with an *IOMMU* (I/O Memory Management Unit) [11], mapping the SG buffer into a contiguous area in the *bus address space* may be possible. However, the versatile DMA engine should not rely on its availability. Additionally, using the *IOMMU* for such translations may result in reduced performance of DMA transfers [12]. Therefore, working with a physically contiguous buffer is preferred because the DMA engine must only know the buffer's start address in the *bus address space* and the buffer's size. On the other hand, the non-contiguous SG buffer must be represented by a list of its contiguous buffers storing their addresses and lengths. The Linux kernel offers a dedicated structure, *sg_table*, to represent such lists.

There are three solutions for working with large DMA buffers with the described limitations. The first two are oriented toward enabling the allocation of the large physically contiguous buffers.

### 1.2.1. Boot Time Reservation

Reserving a large physically contiguous memory area may be performed when booting the operating system. The Linux kernel offers a special *memmap = nn[KMG]$ss[KMG]* parameter that may be used in ACPI-based systems [13]. In the device tree-based systems, a special *reserved-memory* node may be used for that purpose. With the boot-time reservation, however, the user must carefully choose the physical address of the reserved area. Additionally, that memory remains unavailable for the operating system even if the DMA is not used.

### 1.2.2. Contiguous Memory Allocation

As a solution for the above problems, the Contiguous Memory Allocator (CMA) has been proposed [14]. CMA enables moving the pages in memory to consolidate the free pages into larger, physically contiguous areas. It should allow the allocation of large, physically contiguous DMA buffers. However, the CMA is not enabled in standard kernels used in most Linux distributions. The user must recompile the kernel with a modified configuration to use it. Additionally, there is still a risk of unsuccessful CMA allocation in the heavily loaded system.

The third solution aims to enable the DMA engine to work with an SG buffer.

### 1.2.3. Working with Non-Contiguous Buffers

If the DMA engine is supposed to work with a SG buffer, it must be informed about the addresses of all contiguous buffers creating that buffer. Three methods may be used to perform that task (see Figure 3).

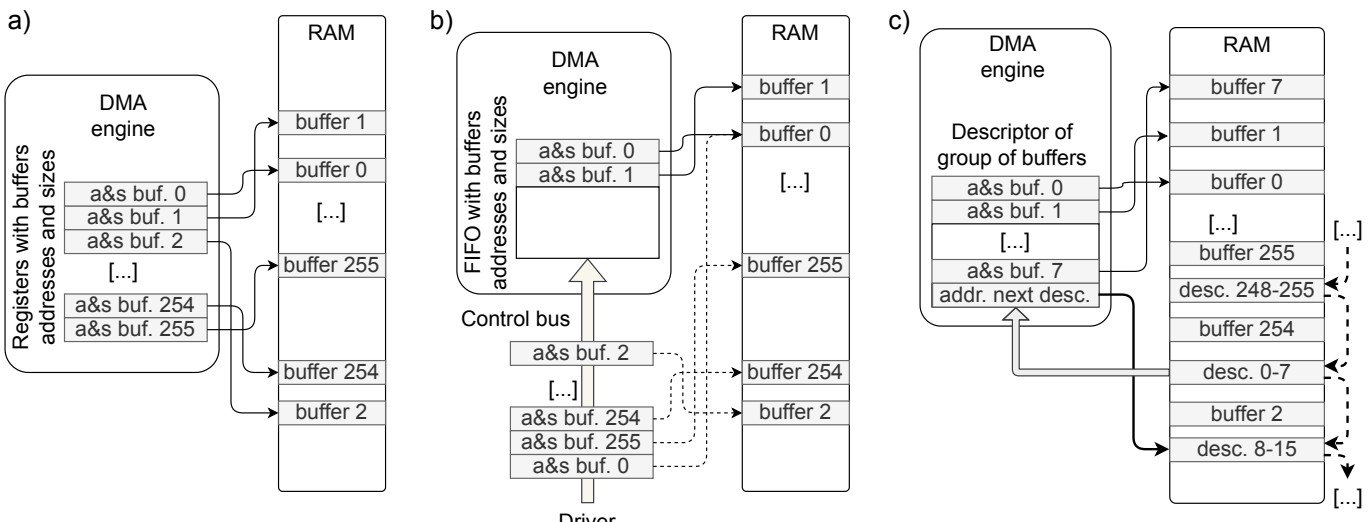

**Figure 3.** Possible DMA designs working with SG buffers. The large SG buffer consists of multiple small, physically contiguous buffers. The example shows the continuous transfer where the SG buffer is used as a circular buffer. In solution (**a**), the DMA engine contains a group of registers storing the addresses of all buffers belonging to the circular buffer. For single-page buffers, it results in huge FPGA resource consumption. In solution (**b**), the addresses of consecutive buffers are cyclically written by the driver to FIFO. This solution increases the CPU load. In solution (**c**), the addresses of groups of buffers are stored in the *descriptors* located in RAM. The DMA engine is given the address of the first descriptor, reads it, and uses the associated buffers. When the end of the group is reached, the next descriptor is used. Such a solution introduces breaks in the data transfer associated with reading the descriptors. It also complicates the data flow in the DMA engine.

In the simplest approach (Figure 3a), the DMA engine may contain a set of registers with a length sufficient to store the data (address and size) of all contiguous buffers being parts of the SG buffer. In the worst case, the buffer may consist of several separate pages

scattered around the RAM. Creating the SG buffer consisting of single pages may even be enforced for simplicity. In that case, we need to store only their physical addresses because they all have the same length. However, with a typical page size of 4 KiB, a 1 GiB SG buffer would require 262,144 pages. Storing so many addresses inside the FPGA would consume too many resources. Therefore, such a solution is unsuitable for big buffers with small pages. The advantage of this solution is that the DMA engine may operate fully autonomously, and it uses the bus only to transfer the data.

The second approach (Figure 3b) requires the device driver to continuously deliver the addresses and sizes of consecutive contiguous buffers. They are stored in FIFO, so the data of the next buffer is available immediately when the current buffer is entirely written. The CPU and bus are additionally loaded, transmitting the buffer's data in this solution.

In the third approach (Figure 3c), the DMA engine reads the information about the consecutive contiguous buffers from the computer memory. For better efficiency, those data are usually stored in descriptors holding the data of a group of buffers (in the figure—for 8 buffers). This solution requires interrupting the transfer whenever the next descriptor must be read. Additionally, handling the descriptors increases the complexity of the DMA engine.

## 2. Existing Solutions for the Implementation of DMA in FPGAs

The implementation of DMA engines in FPGAs is not a new topic. Many solutions have been provided by the FPGA vendors or have been developed independently. Reviewing them all would make this article unacceptably long. Therefore, the review is limited to solutions using AXI-Stream as an input interface and applicable to AMD/Xilinx FPGAs.

### 2.1. Official DMA Engines from AMD/Xilinx

The AMD/Xilinx firm offers many AXI-compatible DMA engines for their FPGAs [15–17]. The deeper analysis shows they are built around the AXI Datamover [18] block. The possibility of using them has been investigated in [19]. The AXI Datamover uses an additional input AXI-Stream interface to receive the transfer commands and another output AXI-Stream interface to send the status of performed transfers. Additional pair of interfaces is used to receive the transferred data and to write it to the target location. Because of that, the concept shown in Figure 3b is a natural way to use it with a SG buffer. An open-source DMA engine based on that concept has been developed, described in [19], and is available in a public git repository [20]. Its block diagram is shown in Figure 4.

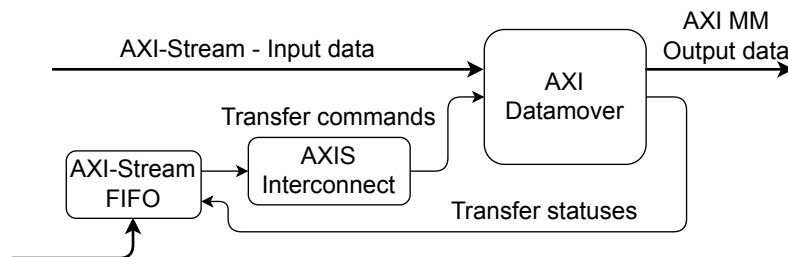

**Figure 4.** Block diagram of the DMA engine based on AXI Datamover. The driver delivers the transfer commands (with data of consecutive contiguous buffers) through the AXI FIFO MM [21] block. The AXI Interconnect block concatenates a few words in a single data transfer command. The statuses of the transfers are sent back to the AXI-Stream FIFO.

The AXI Datamover is continuously provided with the data of consecutive contiguous buffers in advance, which involves the device driver and generates CPU load and traffic on the control bus. When the end of the packet is received, the last buffer may be filled in part. Then, the remaining buffer space is not used because the next packet is stored from the beginning of the next buffer. That results in wasting the buffer capacity. The higher the inefficiency is, the larger the contiguous buffers are. On the other hand, the larger buffers

reduce the load associated with sending their addresses and lengths. In particular, that solution may be highly inefficient for a variable length of packets and a high probability of low-length packets.

AMD/Xilinx also offers a dedicated DMA-capable IP core for PCIe [22]. The usage of its previous version has also been investigated in [19]. It uses the concept of working with SG buffer descriptors stored in the computer system's RAM (as in Figure 3c). Unfortunately, it is a closed-source, proprietary solution limited to operation with the PCIe bus.

### 2.2. Selected Existing Open-Source DMA Engines

From the existing open-source implementations of the DMA engines, three have been selected for this review.

The first is the AXIS2MM DMA engine available in the WB2AXIP [23] suite by Dan Gisselquist [24]. It receives the data from the AXI-Stream, and stores them in a buffer available via an AXI Memory Mapped interface. It supports continuous operation. Unfortunately, it works only with a physically contiguous buffer. The WB2AXIP also contains a SG-capable DMA engine [25], but it only supports memory-to-memory transfers, not stream-to-memory transfers.

Another two open-source DMA engines described in the next section work only with PCIe.

### 2.3. PCIe SG DMA Controller

A "Simple PCIe SG DMA controller" [26] is tightly associated with PCIe. It directly communicates with the TLP layer of the PCIe interface core in FPGA. It is designed for an old FPGA family—Virtex 5. However, it should be easily portable to newer FPGAs. This DMA engine can work in a continuous acquisition mode. It supports SG buffers using method (c) from Figure 3. However, it uses simple descriptors describing only a single buffer. Additionally, it does not work with the AXI-Stream input interface. Possible modification by connecting the AXI-Stream FIFO as an input adapter still would not provide the proper delivery of information about the boundaries between the AXI-Stream packets.

### 2.4. Wupper

Another open-source DMA engine is the Wupper [27]. It has been developed at Nikhef for CERN for the FELIX/ATLAS project. It is a mature and verified in-practice solution. Wupper was intended to be a simple DMA interface for AMD/Xilinx Virtex-7 PCIe Gen 3, but has been ported to newer FPGA families such as Kintex Ultrascale, Kintex Ultrascale+, and Versal Prime. Wupper may work with a few (up to 8) buffer descriptors, with one descriptor always reserved for the transfer from a computer to FPGA. However, according to the documentation, those descriptors' organization does not enable easy handling of SG buffers, especially in the continuous acquisition mode.

## 3. Concept of a Versatile DMA Engine for HEP

Based on the facts described in the introduction and the results of the review of the existing solutions, a concept of a versatile DMA engine for HEP may be formulated.

The engine should be compatible with the SoC/MPSoC using the AXI system bus and with servers using the PCI-Express bus to connect FPGA-based data acquisition cards. Therefore, the engine itself should work as an AXI Master, while a possible connection to the PCIe bus is provided by an additional AXI-to-PCIe bridge (see Figure 1). That solution may be further extended to other buses for which the AXI bridges are available.

The engine should support continuous data acquisition. Hence, it should work with the circular DMA buffer, properly controlling the buffer occupancy and notifying the data processing applications about data availability. Unnecessary use of CPU power should be avoided, and the data notification latency should be minimized.

The engine should work correctly not only in the dedicated data acquisition computer systems, but also in the systems used for development or data processing. It should be easily scalable for different numbers of FPGA boards or different sizes of the DMA buffer. Therefore, the boot-time allocation (see Section 1.2.1) should be avoided.

The maintenance of the system should be simple. Therefore, using a standard Linux distribution should be possible. That eliminates the possibility of using CMA (see Section 1.2.2).

With those limitations, the only remaining option to support large DMA buffers is the scatter-gather (SG) operation (see Section 1.2.3). With the required limiting of the unnecessary CPU load, the best choice is the configuration shown in Figure 3a—storing the data of contiguous buffers in internal registers in FPGA. Unfortunately, for huge and highly fragmented buffers, the number of required registers is enormously high (up to 262,144 registers for a 1 GiB buffer—see Section 1.2.3). The amount of stored information may be reduced by assuming that the buffer consists of single pages. That enables storing only their addresses, as the size is always the same. However, even with that, the resource consumption is unacceptable. Significant improvement is possible by using bigger memory pages. Fortunately, the *x86-64* and 64-bit ARM (*AARCH64*) architectures allow using not only 4 KiB pages, but also 2 MiB ones. Using such "huge pages" reduces the number of required registers by a factor of 512. For example, the 1 GiB DMA buffer consists of 512 single-page contiguous buffers, and their addresses may be easily stored inside FPGA.

Another issue is the efficient communication with the data processing application. The application should be able to sleep while waiting for data availability to reduce the CPU load. The availability of a new complete AXI-Stream packet should generate the interrupt, which via the kernel driver, should wake up the application. However, in case of high intensity in the data stream, it should be possible to mask the interrupt and work in the polling mode (A similar approach is used in Linux drivers for network cards [28]), avoiding wasting CPU time for entering and leaving the interrupt context.

To enable efficient access to the individual packets, the location of the received packets must be available for the receiving application. In addition to the huge circular buffer for data, a smaller circular buffer for packet locations should be created in the computer system (host) memory. A single huge page may be used for that purpose, as shown in Figure 5.

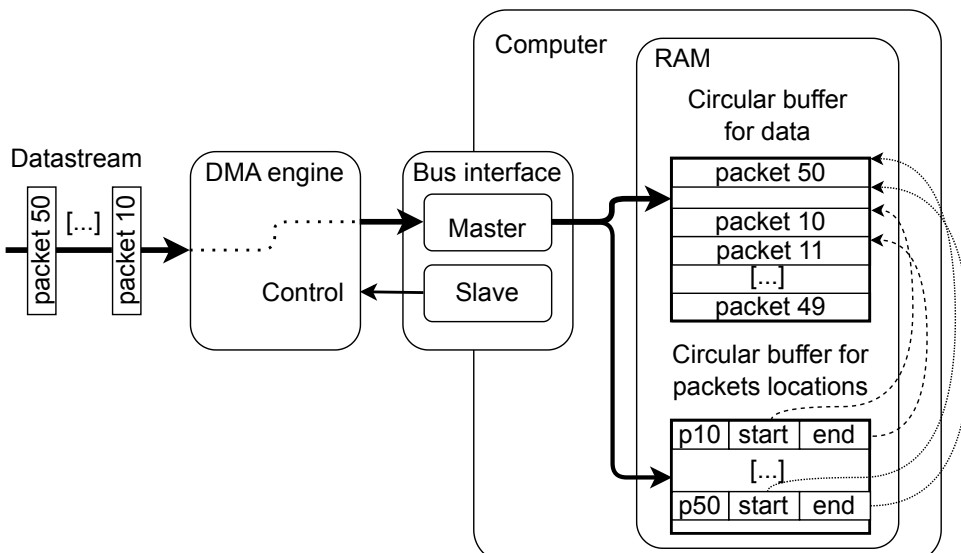

**Figure 5.** Data flow in the DMA engine. The data packets are stored in the primary big circular buffer. The locations of the received packets are stored in a smaller circular buffer, enabling quick access to the desired packet.

## 4. Implementation of DMA in FPGA with HLS

As mentioned in Section 1, the HLS technology may be used to simplify implementing complex algorithms in FPGA. In particular, HLS enables a straightforward (from the user's point of view) implementation of AXI-Stream and full AXI interfaces.

In particular, the implementation of a reasonably performing AXI Master in HDL is quite a sophisticated task [29,30]. In the HLS, simply specifying the appropriate interface for a C/C++ function argument generates a parameterized AXI master [31,32].

A very simple example code from AMD/Xilinx using the AXI Master interface to access the data in the computer system's memory is shown in Listing 1.

**Listing 1.** A very simple code using the AXI interface to read the data from memory and write the modified data to its original location. That is a shortened source published by AMD/Xilinx at [33].

```c
/*
 * Copyright 2021 Xilinx, Inc.
 * Licensed under the Apache License, Version 2.0 (the "License");
*/

#include <stdio.h>
#include <string.h>

void example(volatile int *a){

#pragma HLS INTERFACE m_axi port=a depth=50

  int i;
  int buff[50];

  memcpy(buff,(const int*)a,50*sizeof(int));

  for(i=0; i < 50; i++){
    buff[i] = buff[i] + 100;
  }

  memcpy((int *)a,buff,50*sizeof(int));
}
```

Similarly, specifying the appropriate interface for a C/C++ function argument generates the AXI-Stream slave [31,34].

An implementation of a trivial DMA engine receiving the data from the AXI Stream interface and writing them to the computer system's memory may be performed in less than 70 lines of C/C++ code. An example of such code is published by AMD/Xilinx in [35]. The shortened version is shown in Listing 2.

The function uses two tasks—the first for reading the data from the AXI-Stream to the temporary buffer, and the second for writing the data from that buffer to the computer system's RAM via AXI Master. Both tasks are scheduled using the dataflow approach, allowing them to run in parallel.

The block diagram of that trivial DMA engine is shown in Figure 6.

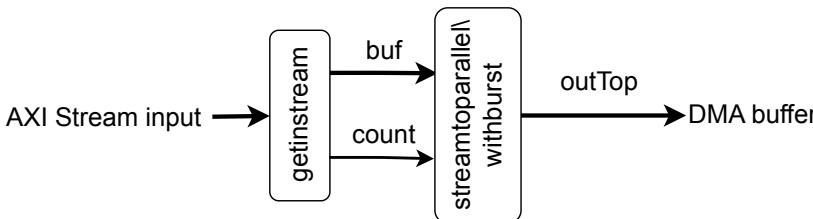

**Figure 6.** Structure of the AMD/Xilinx code implementing the transfer of the AXI-Stream packet to the DMA buffer.

**Listing 2.** Simple code receiving the AXI-Stream packet and storing it in the buffer inside the computer systems memory. That is a modified source published by AMD/Xilinx in [35].

```
/*
 * Copyright 2021 Xilinx, Inc.
 *
 * Licensed under the Apache License,
 * Version 2.0 (the "License");
 */

#include "example.h"

void streamtoparallelwithburst(
      hls::stream<data> &in_stream,
      hls::stream<int> &in_counts,
      ap_uint<64>  *out_memory) {
  data  in_val;
  do {
    int count = in_counts.read();
    for (int i = 0; i < count; ++i) {
#pragma HLS PIPELINE
      in_val = in_stream.read();
      out_memory[i] = in_val.data_filed;
    }
    out_memory += count;
  } while(!in_val.last);
}

void getinstream(
      hls::stream<trans_pkt >& in_stream,
      hls::stream<data > &out_stream,
      hls::stream<int>& out_counts)
{
  int count = 0;
  trans_pkt in_val;
  do {
```

```
#pragma HLS PIPELINE
    in_val = in_stream.read();
    data out_val = {in_val.data, in_val.last};
    out_stream.write(out_val);
    count++;
    if (count >= MAX_BURST_LENGTH || in_val.last) {
      out_counts.write(count);
      count = 0;
    }
  } while(!in_val.last);
}

void example(
      hls::stream<trans_pkt >& inStreamTop,
      ap_uint<64> outTop[1024] ) {
#pragma HLS INTERFACE axis register_mode=both \
  register port=inStreamTop
#pragma HLS INTERFACE m_axi max_write_burst_length=256 \
  latency=10 depth=1024 bundle=gmem0 port=outTop
#pragma HLS INTERFACE s_axilite port = outTop \
  bundle = control
#pragma HLS INTERFACE s_axilite port = return \
  bundle = control

#pragma HLS DATAFLOW

  hls::stream<data,DATA_DEPTH > buf;
  hls::stream<int,COUNT_DEPTH> count;

  getinstream(inStreamTop, buf, count);
  streamtoparallelwithburst(buf, count, outTop);
}
```

### 4.1. Development of the Final HLS Solution

The simple code shown in the previous section does not meet the requirements for the DMA engine for HEP applications, defined in Section 3. It simply reads a single AXI-Stream packet and writes it to the memory buffer. It does not support continuous acquisition nor supports the SG buffers. Additionally, it does not check for overflow in the output buffer.

Adding those necessary functionalities was a long iterative process. The HLS synthesis is controlled with many options, which may be defined as so-called *pragmas* in the source code or as project settings [31,36].

The previous experiences with HLS [37,38] have shown that this process requires thorough verification not only in the C simulation (offered by the Vivado suite), but also at the level of the finally generated RTL code. Therefore, a dedicated verification environment has been created, described in Section 6.1. This section describes the final implementation that uses the HLS kernel with a top-level **dma1** function responsible for handling a single AXI-Stream packet.

The structures and constants used in the implementation are shown in Listing 3. As stated in Section 3, the DMA engine should use the approach shown in Figure 3a with 2 MiB-long huge pages.

The input AXI-Stream interface is implemented as the argument **stin**, while the output AXI Master interface is represented as the argument **a**.

The addresses of the huge pages used as contiguous buffers are described in HLS as the array of 64-bit unsigned integers connected to the AXI Lite interface **bufs**. A constant **NBUFS** defines the maximum number of contiguous buffers, but the argument **nof_bufs** gives their actual number (and, thence, the circular buffer size).

**Listing 3.** Top-level function implementing the core of the DMA engine in HLS. It is described in Section 4.1. The complete source code is publicly available in the repository [39].

```
typedef ap_uint<256> AXI_VALUE;
typedef ap_uint<64> AXI_ADDR;
typedef ap_axiu<256, 1, 1, 1> AXIS_DATA;

typedef struct {
    AXI_VALUE dta;
} BUF_DATA;

typedef struct {
    ap_uint<32> count;
    ap_uint<32> word;
    ap_uint<1> eop;
    ap_uint<1> nextbuf;
} BURST_MARK;

typedef struct {
    ap_uint<64> base;
    ap_uint<64> first;
    ap_uint<64> after;
    ap_uint<32> count;
    ap_uint<32> packet;
    ap_uint<32> nr_buf;
    ap_uint<1> overrun;
    ap_uint<1> eop;
} OUTPUT_CHUNK;

typedef struct {
    ap_uint<32> nr_buf;
    ap_uint<32> nr_pkt;
    ap_uint<1> overrun;
    ap_uint<1> eop;
} OUTPUT_SIGS;

typedef struct {
    ap_uint<64> first;
    ap_uint<64> after;
    ap_uint<64> filler[2];
} PKT_DESC;

static const int BUFFER_FACTOR = 2;

static const int MAX_BURST_LENGTH = 256;

static const int DATA_DEPTH = MAX_BURST_LENGTH * BUFFER_FACTOR;
static const int COUNT_DEPTH = 2*BUFFER_FACTOR;

static const int CHUNKS_DEPTH = 2*BUFFER_FACTOR;
static const int NBUFS = 2048;
#define NPKTS (2*1024*1024/32)  //Number of packets in desc. buffer
#define BUFLEN (2*1024*1024/32)  //Length of buffer in words
```

The address of the huge page storing the packet descriptors (locations of the received packets) is delivered via the argument **descs**.

The next two arguments are used to control filling the circular buffers. The **cur_buf** delivers the number of the first (Because both buffers are circular, the numbers are increased using modular arithmetics) contiguous buffer containing data not yet received by the application. Similarly, the **cur_pkt** delivers the number of the first packet descriptor not yet received by the application.

The arguments **nr_buf** and **nr_pkt** output the number of the contiguous buffer and the number of the packet currently written by the DMA engine.

The last argument, **xoverrun**, outputs the information that the data loss occurred due to an attempt to write new data when either no free contiguous buffer or no free packet descriptor was available.

The top function **dma1** schedules four subtasks (**readin**, **prepare**, **writeout**, and **update_outs**) in the DATAFLOW mode.

Those subtasks are communicating via **hls::stream** variables. The data flow between them is shown in Figure 7, and their functionalities are described in the following subsections.

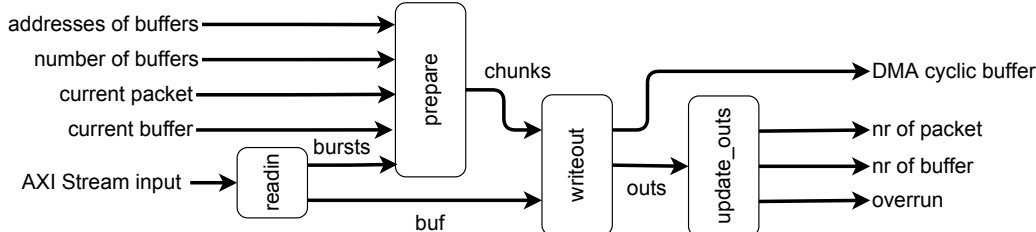

**Figure 7.** Structure of the HLS-implemented core of the DMA engine.

### 4.2. Readin Subtask

This task reads the data from the input AXI-Stream and packs them into "bursts" of the predefined maximum length. The task considers that the data are packed into 2 MiB long contiguous buffers and controls the filling of the contiguous destination buffer. The "burst" is completed when its maximum length is reached, the contiguous destination buffer is filled, or the last word in the AXI-Stream is received. Afterward, the "burst" data

are passed to the **writeout** task. Additionally, the properties are transferred to the **prepare** subtask.

### 4.3. Prepare Subtask

This subtask is given the number of available contiguous buffers and their addresses. It also receives information about the last contiguous buffer and the packet handled and freed by the host. Based on that information and the "burst" properties, this task prepares the descriptor of the write operation (called "chunk") for the next **writeout** task. That descriptor contains:

• A number of the currently transmitted packet,
• A number of the currently used contiguous buffer,
• The destination address and the length of the data,
• The information if overrun (an attempt to write a not freed buffer or packet) occurred,
• The information if the last word from the current AXI-Stream packet was received.

### 4.4. Writeout Subtask

This subtask receives the "chunk" descriptors from the **prepare** task and the associated "burst" data from the **readin** task. Based on that, it writes the "burst" data to the desired location in the host memory. At the end of the AXI-Stream packet, the packet's start and end locations in the circular buffer are also written to the "descriptors" contiguous buffer. Finally, the **writeout** task prepares the new values of the output variables—the number of the currently written contiguous buffer, the number of the currently written data packet (AXI-Stream packet), and the overrun status. Those values are passed to the last subtask **update_outs**.

### 4.5. update_outs Subtask

A dedicated subtask ensures that the output variables are updated after the packet data are successfully stored in the circular data buffer and after the packet descriptor is written to the circular descriptors buffer.

### 4.6. HDL Support Cores

Unfortunately, it was not possible to implement all the required functionality using the HLS technology.

When the driver frees the packet and associated contiguous buffers, the "current buffer" and "current packet" are modified by the software. That change should be immediately visible for the DMA engine. Otherwise, false buffer overruns may be generated.

In the case of interrupt generation, the situation is even worse. The interrupt is generated when the new packet is available, and the interrupts are not masked. The driver keeps the number of the last packet passed to processing in the "last scheduled packet" variable. The availability of the new packet is checked by comparison of the "last scheduled packet" and "nr of packet". After passing the new packet for processing, the software updates the "last scheduled packet". However, when this modification is not immediately visible to the interrupt generation block, a false repeated interrupt request will be generated. For the same reason, all changes in the interrupt masking register must be visible immediately for the interrupt generation block.

Unfortunately, the registers defined in the HLS code as accessible via the **s_axilite** interface do not provide immediate propagation of their values. To workaround the described problems, a separate AXI-Lite slave was implemented in HDL. It provides fast access to the "current buffer", "current packet", "last scheduled packet", and interrupt masking registers, enables writing the HLS core control signals (**ap_start**, **ap_rst_n**) and reading its status (**ap_done**, **ap_ready** and **ap_idle**).



## 5. Software Supporting the DMA Engine

The DMA engine described in the previous section must be supported by the software consisting of the data processing application and the device driver. The device driver creates two separate character devices for each DAQ board available in the system: the **my_daqN** (N is replaced by the number of the board) for the DMA engine and **my_ctrlN** for the DAQ control logic (see Figure 1). Such a solution enables the safe separation of controlling the DAQ system from the data processing. In particular, an error in the data processing thread does not need to crash the DAQ control application (outside this article's scope), so restarting the data acquisition may be possible without full reinitialization of the DAQ system.

The software may work in one of two operating modes. In the single-packet mode, the arriving data packets are processed sequentially. Packets are processed in the order of arrival, and two packets are never processed simultaneously. The application sleeps if there are no more packets for processing. That mode is suitable for debugging, processing the data in a simple single-threaded application, or archiving the data on disk.

In the multi-packet mode, the packets are passed to the data processing threads in the order of arrival. If there is a free processing thread, the next packet may be scheduled for processing before the previous ones are processed. The application sleeps if there are no more packets available for processing. The packet scheduled for processing becomes a property of the processing thread, which is responsible for freeing it after successful processing. That mode enables full utilization of the data processing power of the DAQ host. The packets may be processed in parallel on multiple CPU cores. It is also well suited for transferring data packets independently to the computing grid for processing on different nodes.

The flow diagrams of both modes are shown in Figure 8.

*Detailed Description of the Software Operation*

The software performs the following tasks:

- It prepares the huge pages-backed DMA buffer. It creates a file of the required size in a hugetblfs filesystem (that can be done even in a shell script). Then, the created file is mapped into the application address space.
- Whenever the data acquisition is started or restarted, the following actions must be performed:

    - The DMA driver resets the engine (due to HLS limitations, it is needed to set the initial values of the registers).
    - The DMA driver maps the buffer for DMA (if the buffer was already mapped, the mapping is destroyed and recreated) (This operation requires using functions **get_user_pages_fast** and **__sg_alloc_table_from_pages** or **sg_alloc_table_from_pages_segment**, and the implementation depends on the version of the kernel).
    - The DMA driver configures the DMA engine to work with the currently mapped buffer. In particular, it writes the bus addresses of all huge pages into the engine's registers.
    - The DAQ control application configures the data source.
    - In the multi-packet mode, the data processing application starts the processing threads.
    - The DMA driver starts the engine.
    - The DAQ control application starts the data source.

- If the single-packet mode is used, the data processing loop works as follows:

    - If no data packet is available, the DMA interrupts are switched on, and the application sleeps, waiting for data or command.
    - If the error occurred or the stop command has been received, the application leaves the data processing loop.

- – If the new data packet is received, the DMA interrupts are masked, and the packet is passed to the data processing function.
- – After the packet is processed, it is confirmed and freed.
- – The next iteration of the loop is started.

- If the multi-packet mode is used, the data processing loop works as follows:
  - – If no data packet is available, the DMA interrupts are switched on, and the application sleeps, waiting for data or command.
  - – If the error occurred or the stop command has been received, the application leaves the data processing loop.
  - – If the new data packet is received, its number is passed to one of the data processing threads via ZMQ [40], and the engine is notified that the particular packet has been scheduled for processing. (The device driver uses dedicated **ioctl** commands for that purpose: **DAQ1_IOC_GET_READY_DESC** for obtaining the number of the received packet, and **DAQ1_IOC_CONFIRM_SRV** for writing the number of the last scheduled packet into the **last scheduled packet** register in the engine).
  - – The software checks if other packets have been received and are awaiting processing (A dedicated **ioctl DAQ1_IOC_GET_WRITTEN_DESC** command returns the number of the first packet that is not ready for processing yet. So, all packets between the returned by **DAQ1_IOC_GET_READY_DESC** and that one may be scheduled for processing). In the internal loop, all the available packets are scheduled for processing in the available threads.
  - – The next iteration of the loop is started.

- Actions performed by the signal processing thread in the multi-packet mode are the following:
  - – The thread sleeps, waiting for a packet to be processed.
  - – The parts of the DMA buffer containing the packet descriptors and data of the packet are synchronized for the CPU (A dedicated **ioctl DAQ1_IOC_SYNC** is used for that purpose. Synchronizing the arbitrarily selected part of the SG buffer in the Linux kernel requires storing a separate array of addresses of all huge pages creating the buffer. The original **sg_table** structure does not support random access).
  - – The start and end addresses of the packet data are read from the descriptor.
  - – The packet data are processed.
  - – After the data are processed, the packet is marked for freeing (A dedicated **ioctl DAQ1_IOC_CONFIRM_THAT** command is used for that. Due to the parallel handling of multiple packets, the driver must keep track of all packets ready to be freed. A bitmap is used for that purpose. When the packet currently pointed by the "current packet" register is freed, all the packets marked for freeing are also freed. The "current packet" and "current buffer" are then updated accordingly).
  - – The thread is stopped if the error occurred or the stop command has been received. Otherwise, the above operations are repeated.

- The shutdown procedure:
  - – The DAQ application stops the data source.
  - – The DMA application stops the DMA engine.
  - – In the multi-packet mode, the DMA application sends the STOP command to processing threads and joins them.
  - – The DMA application frees the resources—unmaps, and frees the DMA buffer.

The example data processing application written according to the above description is delivered together with the sources of the driver and is available in the repository [41].

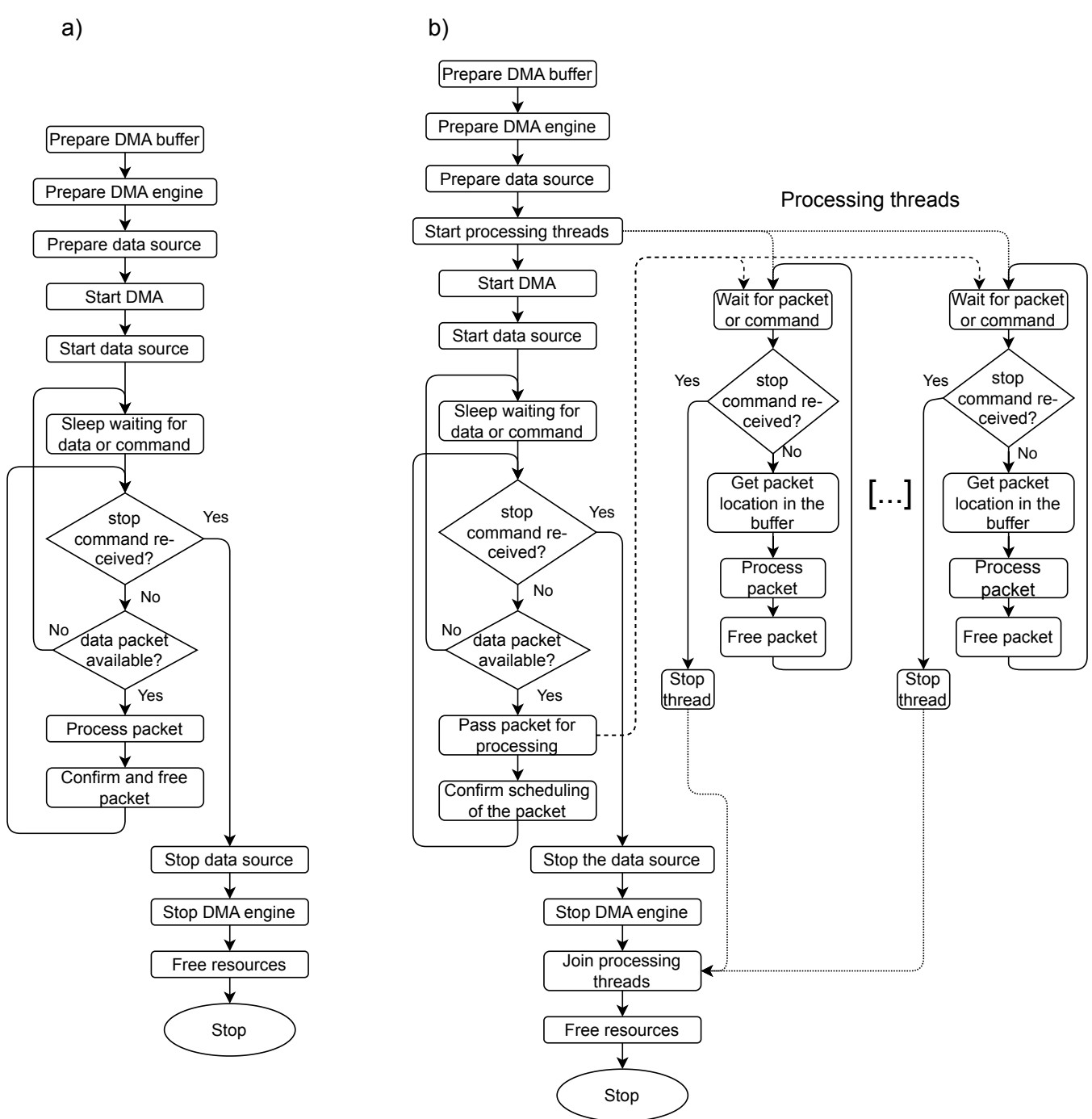

**Figure 8.** Two modes of operation of the data acquisition software; (**a**) single-packet mode—packets are received and processed sequentially, and the next packet is only handled after the previous one is fully processed and confirmed. This mode may be well-suited for recording the acquired data or debugging the system. (**b**) Multi-packet mode—packets are passed for processing to the data processing threads as they arrive. Multiple packets may be processed at the same time. This mode enables full utilization of multiple CPU cores in the system. It may be perfect if different data packets are processed independently (e.g., transferred to different nodes in the computing grid).

## 6. Tests and Results

The DMA engine and accompanying software were thoroughly tested during the development. The initial idea of the core was tested as an entirely virtual device implemented in C in QEMU sources [5]. At this stage, the basic assumptions regarding the architecture,

structure of registers, and driver organization were tested. That emulation environment enabled testing the first version of the driver with different versions of the kernel and different hardware platforms. That contributed to creating the code that compiles on a wide range of kernel versions, starting from 5.4, and works on the x86-64 and AARCH64 architectures used in PCIe-equipped servers and MPSoC systems.

The emulation was maintained throughout the whole development and testing period. The emulated machine could host multiple DMA engines to verify that the driver supports the simultaneous handling of multiple devices. The initial model of the DMA engine was continuously updated to follow the development of the HLS-implemented engine used in the actual hardware. The final version of the model was implemented in two versions— one connected via the PCIe bus [42], and another connected directly to the system bus (emulating the AXI bus) [43]. The driver was slightly modified to support the DMA engine directly connected to the system bus, and the correct operation was confirmed in emulation. Of course, verification in the actual MPSoC system should be done in the future.

### 6.1. Tests in the RTL Simulations

The HLS technology generates the RTL code from the C/C++ description. However, it is a complex process that may be significantly affected even by minor variations of the C/C++ code and the settings used (so-called *pragmas*). Therefore, frequent verification of the generated RTL code was essential to developing the DMA engine in HLS. Complete synthesis and implementation of the generated code take significant time. The capabilities of debugging the core operation in hardware are also limited. The Integrated Logic Analyzer (ILA) [44] allows only a relatively short recording of a preselected subset of internal signals. Modifying this subset requires repeated synthesis and implementation.

Therefore, testing of the generated RTL code was done in HLS simulation. As the PCIe interface simulation is very time-consuming, the simulation was limited to the AXI bus. Verifying the DMA engine working as an AXI Master required a high-performance AXI Slave [45]. Hence, the testbench was created based on the AXI cores developed by Dan Gisselquist for his ZipCPU [23]. Simulation of the computer running the control software also consumes too much time and resources. Therefore it was replaced with a simple controller initializing the DMA core. The block diagram of the main part of the simulation environment is shown in Figure 9.

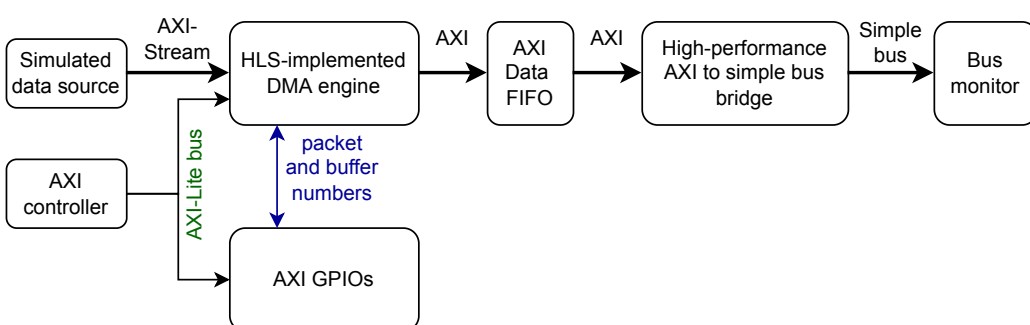

**Figure 9.** The test environment used for simulation. The AXI controller initializes the DMA engine. The high-performance AXI to simple bus bridge was implemented based on the "demofull" core from [23]. The simulated environment is available in the git repository [39] in the *test_env* directory.

The RTL simulations helped to convert the initial simple demonstration code provided by Xilinx (see Listing 2) into the fully-fledged DMA engine working in continuous mode with a large SG buffer (described in Section 4.1 and the following sections).

Those simulations have revealed the problem of fully utilizing the AXI bandwidth. It appeared that the HLS-generated AXI Master does not start sending the next chunk of data before the writing of the previous one is finished and confirmed by the AXI bus. Enabling the generation of outstanding write transactions with the **num_write_outstanding** parameter does not help. The only viable solution was an increase of the chunk length to increase

the ratio of time of writing the chunk to the time of waiting for the confirmation. Of course, such a workaround increases the FPGA memory usage and the data transfer latency. The results of simulations for different chunk lengths are shown in Figures 10 and 11.

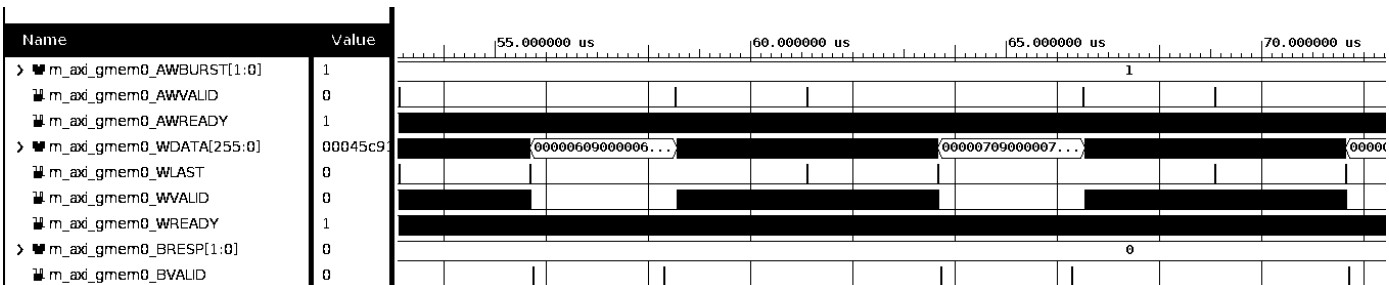

**Figure 10.** Results of simulation for a chunk length of 256 words. Waveform displayed with GTKWave. Approximately 74% of the AXI bus bandwidth is used.

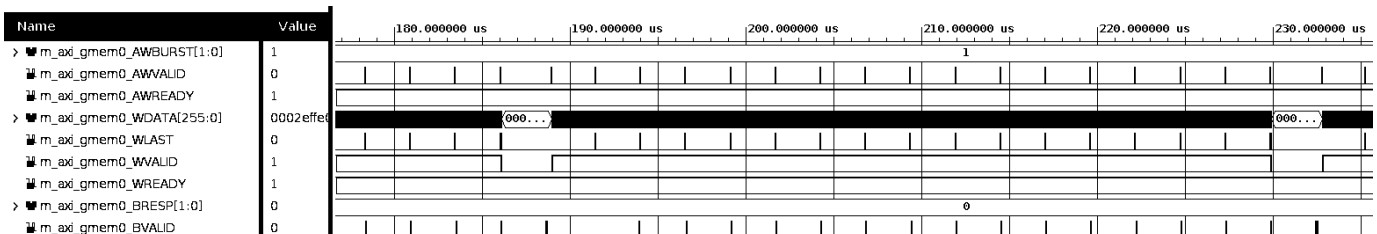

**Figure 11.** Results of simulation for a chunk length of 2048 words. Waveform displayed with GTKWave. Approximately 93% of the AXI bus bandwidth is used.

### 6.2. Tests in the Actual Hardware

The HLS-implemented DMA engine has been successfully synthesized with the Vivado-HLS and Vivado (The DMA engine was prepared for integration with projects using the 2020.1 version of Vivado-HLS and Vivado. Therefore, the same version was used to synthesize, implement and test it) environment for two hardware platforms:

- KCU105 [46] AMD/Xilinx board, equipped with Kintex Ultrascale XCKU040-2FFVA1156E FPGA,
- TEC0330 [47] board from Trenz Electronic equipped with Xilinx Virtex-7 XC7VX330T-2FFG1157C FPGA.

To fully load the board and communication channel, the pseudorandom data from the artificial data generator were transmitted (see Figure 12). The block diagram of the demo project is shown in Figure 12, and the sources are available in the repository [39].

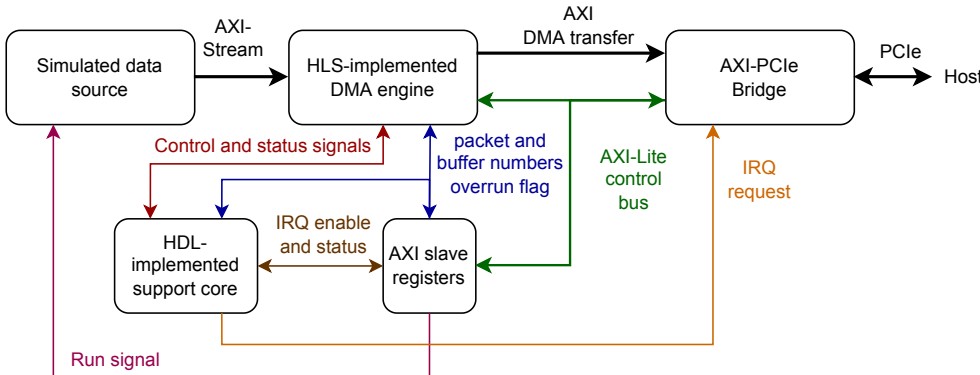

**Figure 12.** Block diagram of the demo project testing the DMA engine with simulated data source in the actual hardware.

The results of synthesis for both platforms and two lengths of the data chunks (Lengths of 256 and 2048 words were used, as in the simulation in Section 6.1, to compare simulated and actual performances) are shown in Table 1. The AXI bus clock frequency was set to 250 MHz as required for an 8xGen3 AXI–PCIe bridge working with a 256-bit wide data bus. The correct timing closure was obtained in all cases.

**Table 1.** Resource consumption of the DMA engine for tested hardware platforms and two lengths of the data chunks. Absolute and percentage (in parenthesis) consumption is given. The artificial data source was included in the design, but the ILA blocks used for debugging were excluded.

| | KCU105 | | | TEC0330 | | |
| | LUTs | Flip Flops | Block RAMs | LUTs | Flip Flops | Block RAMs |
|---|---|---|---|---|---|---|
| Available | 242,400 | 484,800 | 600 | 204,000 | 408,000 | 750 |
| Used for 256-words chunks | 9909 (4.09%) | 15,204 (3.14%) | 45 (7.5%) | 12,503 (6.13%) | 15,928 (3.90%) | 45 (6%) |
| Used for 2048-words chunks | 9858 (4.07%) | 15,213 (3.14%) | 69.5 (11.58%) | 12,445 (6.10%) | 15,946 (3.91%) | 69.5 (9.27%) |

For tests, the KCU105 board was placed in an 8xGen3 PCIe slot of a PC computer with a PRIME B360M-A motherboard, 32 GiB of RAM, and Intel® Core™ i5-9400 CPU @ 2.90 GHz. The TEC0330 board was placed in an 8xGen3 PCIe slot of a Supermicro server with an X10SRi-F motherboard, 64 GiB of RAM, and Intel® Xeon® CPU E5-2630 v3 @ 2.40 GHz.

In both boards, a reliable operation with 8-lanes PCIe Gen 3 was obtained (Trenz Electronic advertises TEC0330 as 8-lanes PCIe Gen 2 capable. However, the FPGA chip used in the board supports PCIe Gen 3, and correct operation in 8xGen3 configuration has been verified in exhaustive tests of three different boards).

The data processing application verified the correctness of all transferred words. The correct operation was confirmed in tests lasting up to 8 h. The firmware compiled for the maximum length of a data chunk equal to 256 words provided low utilization of the available PCIe bandwidth. Therefore, the tests were repeated with the length of the data chunk increased to 2048 words. The results are summarized in Table 2. The measurements of the transfer speed agree with the results obtained in the RTL simulations (see Section 6.1). The bandwidth utilization is lower than in simulations because the PCIe bridge introduces additional latency that delays the write transaction's confirmation. Using the 2048 word-long data chunk provides acceptable performance with reasonable resource consumption. However, the limited utilization of the bus bandwidth requires further investigation.

**Table 2.** Transmission speed and 8xGen3 PCI-Express bandwidth utilization in tested hardware platforms.

| | KCU105 | | TEC0330 | |
| | Absolute | Percentage | Absolute | Percentage |
|---|---|---|---|---|
| Available | 7.877 GB/s | 100% | 7.877 GB/s | 100% |
| Used for 256-words chunks | 4.731 GB/s | 60.1% | 4.721 GB/s | 59.9% |
| Used for 2048-words chunks | 6.724 GB/s | 85.4% | 6.691 GB/s | 84.9% |

## 7. Discussion and Conclusions

The main aim of the work was achieved. A versatile DMA engine was implemented in HLS with minimal supporting HDL code. The accompanying device driver and data-processing application were created. The correct operation of the system in the actual

hardware was confirmed with the PCIe-connected FPGA boards hosted in x86-64 computers. Its correct operation with a FPGA connected directly to the system bus (AXI) was confirmed in simulations.

### 7.1. Innovation and Research Contribution of the Paper

In comparison with the existing solutions described in Section 2, the described system offers the following advantages:

- Using the SG DMA buffer consisting of huge pages and storing the complete description of the huge buffer inside the programmable logic is a new concept not found in the alternative solutions. It significantly simplifies handling the large DMA buffers, which otherwise require one of the below supporting features:
  - Periodically transmitting the addresses of the small contiguous buffers consisting of small standard pages,
  - Reserving the memory at the boot time or using a special version of the Linux kernel with CMA (for allocating a huge physically contiguous buffer),
  - Using a special version of hardware equipped with advanced IOMMU.
- Unlikely the DMA based on the AXI Datamover (see Section 2.1), it fully utilizes the DMA buffer. There is no unused space in the last buffer occupied by the AXI-Stream packet. The start and end position of the packet is stored in the packet descriptors' buffer.
- The presented solution can operate in two hardware configurations: AXI-connected programmable logic (like in MPSoC chips) and PCIe-connected FPGAs. That feature distinguishes it from solutions based on a dedicated DMA-capable IP core for PCIe, a "Simple PCIe SG DMA controller", and Wupper.

A significant contribution of the paper is investigating the applicability of the HLS technology for implementing a DMA engine. The results have shown that HLS may be usable for designing hardware controllers. However, certain functionalities (including real-time handling of control and status signals and interrupt generation) require additional support modules written in HDL. HLS may be used to reduce the workload associated with designing specific parts of the controller (for example, the datapath and AXI interfaces), but cannot fully eliminate the need for an FPGA-skilled engineer.

The described project uses the workflow based on a new design and testing methodology with the device model implemented in QEMU [5]. That approach enabled the development and testing of software components (device driver and data processing application) in conditions that were not possible in the available hardware configurations. For example, testing the simultaneous operation of multiple FPGA boards in the same server or testing the driver in the AARCH-64 platform.

### 7.2. Additional Features Useful in HEP Data Acquisition

The described system offers almost entirely zero-copy operation. The packet data placed into the DMA buffer may be either analyzed locally without copying or passed for transmission to another processing node in the computing grid environment. The system supports the parallel processing of multiple packets, enabling full utilization of the computing power of the host computer.

When used for development, the system's features improve the speed of development and testing. The user may control resources used by the DMA engine without rebooting the computer. Using the huge pages for the SG buffer enables changing the amount of memory allocated for the DMA buffer according to the current needs. The user may decide at the runtime which of the installed DAQ boards should be used. That prevents wasting resources on the installed but unused boards.

### 7.3. Availability of the Sources

The important feature of the presented DMA engine is its open-source character. The sources of all components created by the author are available in public git repositories: the HLS/HDL design in [39], the driver and data processing application in [41], and the QEMU-based emulation environment with the author's models in [42,43]. That is very important because the system, even though usable, still has an experimental character. Therefore, feedback from users is essential.

### 7.4. Practical Use of the System

The developed system has already been used in practice. The TEC0330 board, used as one of the test platforms in this project, is the basis for a GERI(The name GERI is an abbreviation of "GBTxEMU Readout Interface") readout board. The DAQ firmware for GERI was initially developed for the BM@N experiment [48]. Later, it was adapted for other experiments and integrated with the described DMA engine. GERI receives the hit data sent by the STS-XYTER [49] readout ASIC and delivered via the GBTxEMU [50] board. The data received from several (up to 7) GBTxEMU boards are concentrated, supplemented with the metadata describing their source, packed into 256-bit words, and delivered via the AXI-Stream interface to the DMA engine.

The BM@N experiment uses a trigger. Therefore, the data of hits occurring in a predefined period surrounding the trigger pulse is grouped into so-called "event" and transmitted as a single AXI-Stream packet. In other experiments, GERI may be used with a trigger-less readout. In that case, the arriving data are split into the AXI-Stream packets based on their timestamp or time of their arrival.

The firmware was already used by a team preparing the STRASSE detector [51] for the PFAD experiment [52]. The group has achieved the first positive results.

The use of the GERI board with the developed DMA engine is also planned in certain projects related to the CBM experiment [53]. For example, the Indian team working on the MUCH detector [54] for local tests needs GERI to replace a standard CBM readout board, which cannot be used in India due to export restrictions.

Of course, the presented DMA engine may also be used outside the HEP experiments. It may be a useful solution for any FPGA-based application where the fast DMA transfer of data available as AXI-Stream packets is needed.

### 7.5. Future Work

Future work should focus on fixing the discovered deficiencies of the HLS-generated AXI Master to improve the bus bandwidth utilization. If that problem is resolved in newer versions of HLS, porting the design to the newer Vitis-HLS environment may be necessary (Currently, the design requires Vivado and Vivado-HLS 2020.1 because that is the version used by other projects with which it should be integrated).

The engine's operation in FPGA directly connected to the system bus has been verified only in the simulation. Therefore, the implementation of the engine on an MPSoC running Linux should be performed in the near future. Another area for future improvements is integrating the HLS-implemented device with QEMU directly.

**Funding:** This research was partially supported by the statutory funds of Institute of Electronic Systems. This project has also received funding from the European Union's Horizon 2020 research and innovation program under grant agreement No. 871072.

**Data Availability Statement:** Not applicable.

**Acknowledgments:** The author acknowledges support from coworkers from FAIR/GSI. The tests in the actual hardware were partially performed in the FAIR/GSI STS lab, and were supported by Christian J. Schmidt, Jörg Lehnert and David Emschermann.

**Conflicts of Interest:** The author declares no conflict of interest.

## Abbreviations

The following abbreviations are used in this manuscript:

| | |
|---|---|
| FPGA | Field programmable gate array |
| DMA | Direct memory access |
| DAQ | Data acquisition system |
| HEP | High-energy physics |
| SoC | System on chip |
| MPSoC | Multi-processor system on chip |
| TLP | Transaction Layer Packet in the PCI Express interface |
| KiB | 1024 bytes |
| MiB | 1024*1024 bytes |
| GiB | 1024*1024*1024 bytes |

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
