# Peer review of "Versatile DMA Engine for High-Energy Physics Data Acquisition Implemented with High-Level Synthesis"

_electronics, doi:10.3390/electronics12040883_

Round 1

Reviewer 1 Report

This paper has strong points including an open-source implementation, clear descriptions and nice flow.

One of the main concerns is with respect to contribution. The paper presents a "concept" for an adaptable DMA engine. However, the implementation is tied to specific FPGAs. Also, if HLS is doing the hard work, what is the research contribution of this work other than the implementation itself? For example, should all source code be considered research contributions? I was expecting to see a better connection to physics other than only in the introduction. As an FPGA paper, it is a bit weak, because it reads like a lab report. So for the next revision, I would prefer to read what is special about the packets in physics.

There are considerations about the presentation quality of the paper:

- The figures and algorithms in the paper are a bit wasteful. The block design diagram and the Xilinx templates are usually excluded from scientific papers, as they are redundant etc. For example, figure 9 can be redrawn from scratch as a smaller figure (e.g. notice how the font is too small).

- Grammar issues: "accompanying kernel driver and user space data 522
application were created. Correct operation"

- Table 1 is outside the boundaries of the manuscript

- Figures 10 and 11 can be redrawn with Latex etc.

- Frequent variable names in the text "axctl_0" make it read like a lab report rather than a research publication.

- The maximal operating frequency must be mentioned

- x86_64 -> x86-64

- The references mainly consist of repositories etc. which also reflect the aforementioned problem about source code itself being a research contribution

In summary, this paper reads well, but it requires a related work (sub)section (in connection with research), better definition of the contributions as a research work, a better connection to physics, use the figure/table/algorithm space more effectively and avoid tool screenshots.

Author Response

Dear Reviewer, thank you for your review. The response is in the atteached PDF file.

Reviewer 2 Report

In this paper, the authors present a versatile DMA engine implemented in HLS, that can be used in systems with FPGA-equipped PCIe boards hosted in a server and MPSoC-based systems with programmable logic connected directly to the AXI system bus. The paper deals with an interesting topic, however,

- The paper should be more concise, clear and focused on the contribution because in its actual form it has important problems of organization and length that authors should change removing a lot of unimportant content.

- In the introduction part, show what the originality of your work is.

- The study lacks a clear comparison between the submitted paper and the more relevant literature contributions, which should highlight the main advantages of the current submission.

- I encourage the authors to have their manuscript proof-edited by a native English speaker to enhance paper presentation levels. 

Author Response

(The authors gave the same response as above.)
